# The price of ignorance: how much does it cost to forget noise structure in low-rank matrix estimation?

**Jean Barbier**[*]
International Center for Theoretical Physics
jbarbier@ictp.it

**Tianqi Hou**
Huawei
thou@connect.ust.hk

**Marco Mondelli**[*]
Institute of Science and Technology Austria
marco.mondelli@ist.ac.at

**Manuel Sáenz**[*]
International Center for Theoretical Physics
msaenz@ictp.it

## Abstract

We consider the problem of estimating a rank-1 signal corrupted by structured rotationally invariant noise, and address the following question: *how well do inference algorithms perform when the noise statistics is unknown and hence Gaussian noise is assumed?* While the matched Bayes-optimal setting with unstructured noise is well understood, the analysis of this mismatched problem is only at its premises. In this paper, we make a step towards understanding the effect of the strong source of mismatch which is the noise statistics. Our main technical contribution is the rigorous analysis of a Bayes estimator and of an approximate message passing (AMP) algorithm, both of which incorrectly assume a Gaussian setup. The first result exploits the theory of spherical integrals and of low-rank matrix perturbations; the idea behind the second one is to design and analyze an artificial AMP which, by taking advantage of the flexibility in the denoisers, is able to "correct" the mismatch. Armed with these sharp asymptotic characterizations, we unveil a rich and often unexpected phenomenology. For example, despite AMP is in principle designed to efficiently compute the Bayes estimator, the former is *outperformed* by the latter in terms of mean-square error. We show that this performance gap is due to an incorrect estimation of the signal norm. In fact, when the SNR is large enough, the overlaps of the AMP and the Bayes estimator coincide, and they even match those of optimal estimators taking into account the structure of the noise.

## 1 Introduction

The estimation of a low-rank matrix from noisy data is a central problem in machine learning, and it appears, e.g., in sparse principal component analysis (PCA) [50, 88], community detection [1, 62], and group synchronization [69]. In this paper, we consider the prototypical task of recovering a symmetric rank-1 matrix $\boldsymbol{X}\boldsymbol{X}^\top$ from noisy observations of the form

$$\boldsymbol{Y} = \frac{\sqrt{\lambda_*}}{N}\boldsymbol{X}\boldsymbol{X}^\top + \boldsymbol{Z} \in \mathbb{R}^{N \times N}. \tag{1}$$

Here, $\lambda_* > 0$ is the signal-to-noise ratio (SNR) which quantifies the signal strength, and $\boldsymbol{Z} \in \mathbb{R}^{N \times N}$ is a random matrix that captures the noise. A natural estimator of $\boldsymbol{X}$ is given by the principal eigenvector of $\boldsymbol{Y}$. Its performance and, more generally, the behavior of the eigenvalues and eigenvectors of (1) has been studied in exquisite detail in statistics [49, 68] and random matrix theory [7, 8, 21, 22, 26, 40, 52, 6]. Going beyond the spectral estimator given by the principal eigenvector, approximate message

36th Conference on Neural Information Processing Systems (NeurIPS 2022).

---

[*]Equal contribution

passing (AMP) constitutes a popular family of iterative algorithms. The reason for this popularity lies in two especially attractive features: *(i)* AMP algorithms can be tailored to exploit knowledge on the structure of the signal; and *(ii)* under suitable assumptions, their performance in the high-dimensional limit is accurately tracked by a deterministic recursion known as *state evolution* [18, 24, 48]. Using the state evolution machinery, it has been showed that AMP achieves Bayes-optimal performance in some Gaussian models [29, 30, 61, 13], and a bold conjecture from statistical physics is that AMP is optimal in the class of polynomial-time algorithms for a large class of inference problems with random design. AMP algorithms have been recently considered for several practical scenarios, including genetics [86], inpainting [66] and MRI image recovery [76].

A general class of noise models that has received attention in the literature is given by the family of *rotationally invariant matrices*. This is much milder than requiring $Z$ to be Gaussian: it only imposes that the matrix of eigenvectors is uniformly random, and allows for an arbitrary spectrum. Hence, $Z$ can capture the complex correlation structure which often occurs in applications (e.g., recommender systems [20] and bioinformatics [72]). However, estimating noise statistics from data is costly or, for problems involving large-scale datasets (computational genomics is a paradigmatic example [36, 2]), may be impossible. Thus, one natural idea is to simply assume Gaussian statistics for $Z$. The case in which $Z$ is actually Gaussian has been thoroughly studied [53, 28, 55, 54, 12, 61]. Beyond Gaussianity, a rapidly growing literature is focusing on rotationally invariant models assuming *perfect knowledge of the statistics of the structured matrix appearing in the problem* (such as noise in inference, a sensing, data, or coding matrix in regression tasks, weight matrices in neural networks, or a matrix of interactions in spin glass models) [67, 23, 37, 38, 42, 65, 56, 57, 74, 78, 79, 16, 33, 66, 71, 47]. However, despite this impressive progress when the noise statistics is known, low-rank estimation in a *mismatched setting with partial to no knowledge of the statistics of the rotationally invariant noise matrix remains poorly understood*. The goal of this work is thus to shed light into the following fundamental question:

*Suppose that the noise statistics is unknown or unreacheable, and hence Gaussian noise is naively assumed. What is the impact of this mismatch on the overall performance of inference methods?*

## 1.1 Summary of contributions

In this paper, we provide rigorous performance guarantees for two inference methods which incorrectly assume Gaussian noise statistics: a Bayes estimator, and an AMP algorithm. Then, by exploiting these sharp analytical characterizations, we describe a number of surprising effects coming from numerical simulations. Our main findings are detailed below.

**Theoretical results.** *(i)* We give a closed-form expression for the mean-square error (MSE) of the Bayes estimator which samples from the *mismatched* posterior (cf. Theorem 1). Under a certain concentration assumption, we also present an asymptotic result on the overlap of such estimator. *(ii)* We provide a state evolution analysis (cf. Theorem 2) for the *Gaussian* AMP algorithm which is designed for Gaussian noise.

**Numerical results.** The mismatched Bayes and AMP estimators display surprising behaviors – already for the case of a spherically distributed signal – and the two performance metrics (MSE and overlap) exhibit a remarkably different phenomenology: *(i)* As for the MSE, the Gaussian AMP is *outperformed* by the Bayes estimator. Here, the surprise comes from the fact that AMP algorithms are, in principle, designed to sample from the posterior distribution (and this is often what happens in the matched case). *(ii)* In contrast, when the SNR is large enough, the overlaps of the two mismatched estimators *coincide*, and they even match the overlap of estimators which exploit the noise statistics, namely the optimal spectral method which minimizes the MSE and the correct AMP designed in [37]. *(iii)* Under certain conditions, the MSE of the mismatched Bayes estimator *matches* that of a Gaussian spectral method with no information on the noise structure. *(iv)* The mismatched estimators are *outperformed* – in terms of MSE – by the optimal spectral method and the correct AMP, whose performance coincide. *(v)* Finally, the MSE curves of the Bayes and Gaussian spectral methods exhibit a striking non-monotone behavior.

## 1.2 Related work

**The impact of mismatch.** Given the practical relevance of understanding the effect of mismatch in statistical inference, a line of work has approached the issue from various angles. Regression being

probably the most paradigmatic model of inference task, mismatch has been thoroughly studied in this context. In information theory, the trend was initiated in [83, 85]. Studies for M-estimation and robust statistics followed based on statistical mechanics approaches [3], on the analysis of approximate message passing algorithms [25, 77, 32, 43], on Gordon's convex min-max theorem [81, 80] and the leave-one-out method [35, 34]. Concerning the analysis of Bayesian approaches to mismatched linear regression, we refer to [27, 63, 75, 11] or [9] for a review.

In contrast, the problem studied in the present paper, namely low-rank matrix inference with mismatch, has received attention only recently. It is clear that in some way, related questions were analyzed in the aforementioned random matrix theory literature, although from a rather different perspective. Concerning the precise issue of the performance degradation due to mismatch in Bayesian inference, an exception (and inspiration for our work) is the recent paper [71]. However, the authors of [71] focus on mismatch in the signal-to-noise ratio for the Gaussian noise setting; their results can thus be recovered as a special case of the present ones. To the best of our knowledge, this is the first work that considers mismatch in the noise statistics for low-rank matrix estimation.

**Approximate message passing.** AMP algorithms have been applied to a wide range of inference problems. Examples include estimation in linear models [19, 18, 31], generalized linear models [73, 13, 58, 59], and low-rank matrix recovery with Gaussian noise [17, 29, 41, 51, 55, 61], see also the survey [39]. A general AMP iteration for rotationally invariant matrices has been recently analyzed in [37, 87], and by providing suitable instances of this abstract iteration, AMP algorithms have been developed for low-rank [37, 87, 60] and generalized linear models [82]. Furthermore, an AMP-based method which uses the classical idea of empirical Bayes to reduce the high-dimensional noise in PCA is proposed in [86], which also provides applications to genetics. However, the existing results cannot be applied to the mismatched setting considered in this work, since the Gaussian AMP does not contain the right Onsager corrections. In fact, the algorithm designer assumes Gaussian statistics for the noise and, therefore, constructs an AMP algorithm with the Onsager correction suitable for Gaussian noise. Finally, we remark that the performance of the Gaussian AMP is numerically compared with that of the correct AMP (exploiting the knowledge of noise statistics) in [87]. Our Theorem 2 provides rigorous foundations for such a comparison.

## 2 Setup of the problem

### 2.1 Random matrix theory preliminaries

We start with some useful notions of random matrix theory. Given a probability measure $\rho$ of compact support $K \subseteq \mathbb{R}$, we let $H_\rho : \mathbb{R} \backslash K \mapsto \mathbb{R}$ be the *Hilbert transform* of $\rho$: $H_\rho(z) := \int_K \rho(d\gamma)(z-\gamma)^{-1}$. We define $\overline{\gamma} := \max K$, $\underline{\gamma} := \min K$, $\overline{h} := \lim_{z \downarrow \overline{\gamma}} H_\rho(z)$ and $\underline{h} := \lim_{z \uparrow \underline{\gamma}} H_\rho(z)$, where the limits exist due to the monotonicity of $H_\rho$ but may be infinite. As $H_\rho$ is a bijection between $\mathbb{R} \backslash K$ and its image $(\underline{h}, \overline{h}) \backslash \{0\}$, its inverse exists and it is denoted by $K_\rho : (\underline{h}, \overline{h}) \backslash \{0\} \mapsto \mathbb{R} \backslash K$. The *R-transform* of $\rho$ is $R_\rho : (\underline{h}, \overline{h}) \backslash \{0\} \mapsto \mathbb{R} \backslash K$ given by $R_\rho(x) := K_\rho(x) - 1/x$. The coefficients $\{\overline{\kappa}_k\}_{k \geq 1}$ of the Taylor series of $R_\rho$ are the *free cumulants* associated to $\rho$, i.e., $R_\rho(x) = \sum_{k=0}^{\infty} \overline{\kappa}_{k+1} x^k$, and they can be computed from the moments of $\rho$, see e.g. Section 2.5 of [64]. Furthermore, $R'_\rho(x)$ and $H'_\rho(z)$ denote the derivatives of the $R$-transform and the Hilbert transform of $\rho$, respectively.

### 2.2 Model of mismatched low-rank matrix estimation

We consider the problem of estimating a rank-one informative *spike* $\boldsymbol{XX}^\top$ corrupted by an additive symmetric noise matrix $\boldsymbol{Z} \in \mathbb{R}^{N \times N}$ from data $\boldsymbol{Y}$ generated as in (1). The scalar $\lambda_* \in \mathbb{R}_{\geq 0}$ is the signal-to-noise ratio (SNR), and the noise $\boldsymbol{Z}$ can be decomposed as $\boldsymbol{Z} = \boldsymbol{O\Sigma O}^\top$, where $\boldsymbol{\Sigma} := \text{diag}(\gamma_1, \dots, \gamma_N) \in \mathbb{R}^{N \times N}$ is a diagonal matrix containing the eigenvalues of $\boldsymbol{Z}$ and $\boldsymbol{O}$ is some orthogonal matrix. We also denote by $\overline{\gamma}_N$ and $\underline{\gamma}_N$ respectively $\max\{\gamma_1, \dots, \gamma_N\}$ and $\min\{\gamma_1, \dots, \gamma_N\}$. Similarly, $\overline{\nu}_N$ and $\underline{\nu}_N$ denote the largest and smallest eigenvalues of the data matrix $\boldsymbol{Y}$. Throughout the paper, the main technical assumption will be the following.

**Assumption 1.** *The signal $\boldsymbol{X}$ has norm $\sqrt{N}$. The random noise matrix $\boldsymbol{Z}$ is rotationally invariant and is independent of $\boldsymbol{X}$. Moreover, the empirical measure $N^{-1} \sum_{i \leq N} \delta_{\gamma_i}$ of eigenvalues of $\boldsymbol{Z}$ converges weakly towards a limiting measure $\rho$ of compact support $K \subseteq \mathbb{R}$. Finally, $\overline{\gamma}_N$ and $\underline{\gamma}_N$ converge a.s. to $\overline{\gamma} = \max K$ and $\underline{\gamma} = \min K$, respectively.*

The noise matrix $\boldsymbol{Z}$ being rotationally invariant means that $\boldsymbol{O}$ is a random orthogonal matrix (i.e., sampled from the Haar measure) independent of $\boldsymbol{\Sigma}$. Under Assumption 1, we have that $\bar{\nu}_N$ and $\underline{\nu}_N$ converge a.s. to finite limits, which we denote by $\bar{\nu}$ and $\underline{\nu}$, respectively. These have an explicit form in terms of $\rho$ and $\lambda_*$ given in [21, Theorem 2.1] and which we reproduce for convenience in Appendix A. We remark that the assumption on the ground-truth signal is rather mild: $\boldsymbol{X}$ can have any distribution over the sphere of radius $\sqrt{N}$, and it might even be deterministic. For example, $\boldsymbol{X} \in \{-1, 1\}^N$ could be a uniform vector over the centred $N$-dimensional hyper-cube of sides $2\sqrt{N}$.

For the analysis of the Bayes estimator, we require a second assumption on the asymptotic eigenvalue distribution $\rho$. This ensures that the limit for the overlap of the spectral estimators with the signal is continuous in the SNR, whenever a small additive Wigner noise of variance $\epsilon \geq 0$ is added to $\boldsymbol{Y}$.

**Assumption 2.** *Let $\rho_\epsilon$ be the spectral density of the free convolution of $\rho$ and a semicircle law of radius $2\epsilon \geq 0$. Let $H_{\rho_\epsilon}$ be the associated Hilbert transform and $\bar{\gamma}_\epsilon$ the rightmost point of the support of $\rho_\epsilon$. Then, we assume that $\lim_{z \downarrow \bar{\gamma}_\epsilon} H'_{\rho_\epsilon}(z) = -\infty$ for all $\epsilon \geq 0$.*

For the definition of the free convolution and its link to random matrix theory, we refer the reader to [4, 70]. We remark that Assumption 2 is satisfied by a wide class of random matrices. In particular, by combining Theorem 2.2 of [10] with Proposition 2.4 of [21], it suffices that the support of the limiting spectral measure $\rho$ is *(i)* compact, *(ii)* connected, and *(iii)* it has a proper decay rate at its edges (i.e., $\rho$ is of Jacobi type), see Assumption 2.1 of [10]. These conditions can be easily checked for many random matrix models, including the ones discussed in the examples of Section 4.

Note that, if $\rho$ is the semicircle law of radius 2, the noise is asymptotically equal in distribution to a standard Wigner matrix $\boldsymbol{W}$ with density $\sim \exp(-\frac{N}{4}\mathrm{Tr}\boldsymbol{W}^2)$. As the elements of this type of symmetric matrices are i.i.d. Gaussian, in this case we say that there is *Gaussian noise*. For other limiting $\rho$, the elements of $\boldsymbol{Z} \neq \boldsymbol{W}$ remain correlated and we say that there is *structured noise*. We denote by $\boldsymbol{W}$ a sequence of standard Wigner matrices while we reserve $\boldsymbol{Z}$ for a generic sequence of rotationally invariant matrices.

**Sources of mismatch.**   In this paper, we consider what happens when there is a *mismatch* between the true noise statistics and the assumptions on the noise statistics made in the inference algorithm. In particular, we study the case in which the noise is assumed to be Gaussian. Gaussianity is in fact the most standard assumption made when precise knowledge of the noise structure is lacking. We also consider the case in which the SNR is estimated incorrectly, i.e., the statistician assumes that the data is generated according to $(\sqrt{\lambda}/N)\boldsymbol{X}\boldsymbol{X}^\top + \boldsymbol{W}$, where $\lambda \neq \lambda_*$ and $\lambda_*$ is the correct SNR present in (1). Our goal is to quantify how these sources of ignorance affect the algorithmic performance.

### 2.3   Three classes of inference procedures for PCA

**Mismatched Bayes estimator.**   The statistician assumes Gaussian noise and that the signal $\boldsymbol{X}$ has no specific structure, i.e., it is uniformly distributed on the $N$-dimensional sphere $\mathbb{S}^{N-1}(\sqrt{N})$. The mismatched posterior distribution reads, using the Gaussian log-likelihood $-\frac{N}{4}\mathrm{Tr}(\boldsymbol{Y} - \frac{\sqrt{\lambda}}{N}\boldsymbol{x}\boldsymbol{x}^\top)^2$,

$$P_{\mathrm{mis}}(d\boldsymbol{x} \mid \boldsymbol{Y}) = \frac{1}{Z_N(\boldsymbol{Y})} \exp\Big(\frac{\sqrt{\lambda}}{2}\langle \boldsymbol{x}, \boldsymbol{Y}\boldsymbol{x}\rangle\Big)\mu_N(d\boldsymbol{x}), \tag{2}$$

where $Z_N(\boldsymbol{Y})$ is the normalization constant and $\mu_N$ is the uniform measure over $\mathbb{S}^{N-1}(\sqrt{N})$. In case the signal $\boldsymbol{X}$ lies on the sphere (by Assumption 1) but is not uniformly distributed on it, then this leads to a third source of mismatch. However, the MSE formula of Theorem 1 does *not* depend on it; e.g. if $\boldsymbol{X} \in \{-1, 1\}^N$ the error remains the same. The origin of this is the uniformly spherical prior used in the posterior (2) which is the most uninformative one.

The Bayes estimators we analyze are obtained as the posterior means

$$M_{\mathrm{mis}}(\boldsymbol{Y}) := \int \boldsymbol{x}\boldsymbol{x}^\top P_{\mathrm{mis}}(d\boldsymbol{x} \mid \boldsymbol{Y}), \tag{3}$$

which may not be practical to compute. Notice that, if the noise is Wigner, then the likelihood $\exp(\cdots)$ in (2) is correct. This Bayes-optimal case has already been rigorously characterized (see, for example, [12, 14, 54]), and it will serve us as a base case for comparison with our mismatched setting.

The performance of a sequence of estimators $M_N(\boldsymbol{Y}) \in \mathbb{R}^{N \times N}$ of the spike $\boldsymbol{X}\boldsymbol{X}^\top$ is quantified via the asymptotic mean-square error (MSE):

$$\text{MSE}(M) \coloneqq \lim_{N \to +\infty} \frac{1}{2N^2} \mathbb{E}\|\boldsymbol{X}\boldsymbol{X}^\top - M_N(\boldsymbol{Y})\|_{\text{F}}^2, \tag{4}$$

where $\|\cdot\|_{\text{F}}^2$ denotes the Frobenius norm and $\mathbb{E}$ the expectation over the spike $\boldsymbol{X}\boldsymbol{X}^\top$ and the noise $\boldsymbol{Z}$. We also consider another performance measure which is insensitive to the norm of an estimator $\hat{\boldsymbol{x}}$ of $\boldsymbol{X}$, namely, the rescaled overlap:

$$\text{Overlap}(\hat{\boldsymbol{x}}) \coloneqq \lim_{N \to +\infty} \frac{|\langle \boldsymbol{X}, \hat{\boldsymbol{x}}(\boldsymbol{Y})\rangle|}{\|\hat{\boldsymbol{x}}(\boldsymbol{Y})\| \cdot \|\boldsymbol{X}\|}, \tag{5}$$

where $\|\boldsymbol{X}\| = \sqrt{N}$ due to the spherical constraint. This alignment measure is sometimes also referred to as *cosine distance*. For both AMP and spectral estimators, this overlap will almost surely be deterministic. If the norm of the estimator vanishes, by convention we fix the overlap to zero. Note that the definition (5) is not meaningful for the Bayes case. In fact, if one defines $\hat{\boldsymbol{x}}_{\text{mis}} \coloneqq \int \boldsymbol{x} P_{\text{mis}}(d\boldsymbol{x} \mid \boldsymbol{Y})$, then this quantity is zero by sign symmetry ($\pm\boldsymbol{x}$ have the same posterior weight). Thus, the correct definition of the overlap for the mismatched Bayes estimator is

$$\text{Overlap}_{\text{mis}} \coloneqq \lim_{N \to +\infty} \left( \frac{1}{N} \frac{1}{\|M_{\text{mis}}(\boldsymbol{Y})\|_{\text{F}}} \int P_{\text{mis}}(d\boldsymbol{x} \mid \boldsymbol{Y})\langle \boldsymbol{X}, \boldsymbol{x}\rangle^2 \right)^{1/2}, \tag{6}$$

while for the "one-shot" AMP and spectral estimators we use the former definition (5).

**Approximate message passing.** We remark that, as opposed to our analysis of the Bayes estimator, the state evolution characterization of AMP (Theorem 2) does not require $\boldsymbol{X}$ to be uniformly distributed on $\mathbb{S}^{N-1}(\sqrt{N})$. The AMP analysis can accomodate more generic (potentially mismatched) prior distributions and, in fact, AMP algorithms are well equipped to exploit structure in the signal. For simplicity, we assume to have access to an initialization $\hat{\boldsymbol{x}}^1 \in \mathbb{R}^N$, which is independent of the noise $\boldsymbol{Z}$ and has a strictly positive correlation with $\boldsymbol{X}$, i.e.,

$$(\boldsymbol{X}, \hat{\boldsymbol{x}}^1) \xrightarrow{W_2} (X, \hat{x}_1), \quad \mathbb{E}[X\,\hat{x}_1] \coloneqq \epsilon > 0, \quad \mathbb{E}[\hat{x}_1^2] \le 1, \tag{7}$$

where $(\boldsymbol{X}, \hat{\boldsymbol{x}}^1) \xrightarrow{W_2} (X, \hat{x}_1)$ denotes convergence of the joint empirical distribution of $(\boldsymbol{X}, \hat{\boldsymbol{x}}^1)$ to the random variable $(X, \hat{x}_1)$ in Wasserstein-2 ($W_2$) distance. Then, for $t \ge 1$ the AMP iteration reads

$$\boldsymbol{x}^t = \boldsymbol{Y}\hat{\boldsymbol{x}}^t - \beta_t\hat{\boldsymbol{x}}^{t-1}, \quad \hat{\boldsymbol{x}}^{t+1} = h_{t+1}(\boldsymbol{x}^t), \tag{8}$$

where we assume that $\hat{\boldsymbol{x}}^0 = \boldsymbol{0}$. Here, the function $h_{t+1} : \mathbb{R} \to \mathbb{R}$ is applied component-wise, and it can be chosen in order to exploit prior information about the signal; $\beta_1 = 0$ and, for $t \ge 2$, $\beta_t = \langle h'_t(\boldsymbol{x}^{t-1})\rangle$, where $h'_t$ denotes the derivative of $h_t$. The AMP estimator of $\boldsymbol{X}$ is $\hat{\boldsymbol{x}}^t$, and the one of the spike $\boldsymbol{X}\boldsymbol{X}^\top$ is $M_{\text{AMP}}^t = \hat{\boldsymbol{x}}^t(\hat{\boldsymbol{x}}^t)^\top$. We refer to this algorithm as *Gaussian AMP*, since this is the AMP that is implemented for Gaussian noise (and known SNR). For a discussion on how (8) can be derived, we refer the interested reader to the review [39]. We note that the initialization (7) is impractical and one can design AMP iterations which are initialized with the eigenvector $\overline{\boldsymbol{v}}_N$ of the data matrix $\boldsymbol{Y}$ associated to the largest eigenvalue $\overline{\nu}_N$, see [61, 60, 87]. We are also able to provide a state evolution result for the Gaussian AMP with spectral initialization, and this is discussed in Appendix C.4.

**Spectral estimators.** Finally, we consider spectral estimators of the form $C\,\overline{\boldsymbol{v}}_N\overline{\boldsymbol{v}}_N^\top$, where $\overline{\boldsymbol{v}}_N$ is the unit-norm eigenvector associated to the largest eigenvalue $\overline{\nu}_N(\boldsymbol{Y})$ and $C > 0$ is a scaling constant taking into account the spectrum of the data matrix. The asymptotic description of $\overline{\boldsymbol{v}}_N$ and $\overline{\nu}_N$ for additive and multiplicative low-rank perturbations of rotationally invariant matrices has been obtained in [21, 22, 6]. In particular, by Theorem 2.2 of [22], the squared overlap $\langle \boldsymbol{X}, \overline{\boldsymbol{v}}_N\rangle^2/N$ converges a.s. to a constant $C(\rho, \lambda_*)$, whose explicit expression is recalled in Appendix A. We study two variants:

• The *optimal spectral estimator (OptSpec)* is given by $M_{\text{OS}} \coloneqq C_{\text{OS}}\overline{\boldsymbol{v}}_N\overline{\boldsymbol{v}}_N^\top$, where $C_{\text{OS}}(\rho, \lambda_*) \coloneqq C(\rho, \lambda_*)$ is the optimal scaling that minimizes the MSE.

• The *Gaussian mismatched spectral estimator (GauSpec)* is given by $M_{\text{GS}} \coloneqq C_{\text{GS}}\overline{\boldsymbol{v}}_N\overline{\boldsymbol{v}}_N^\top$, where $C_{\text{GS}}$ is the optimal scaling if the noise was Gaussian and the SNR was equal to $\lambda > 0$. By letting $\rho_{\text{SC}}$ be the standard semi-circle law, we have $C_{\text{GS}}(\lambda) \coloneqq C(\rho_{\text{SC}}, \lambda) = 1 - 1/\lambda$.

For both OptSpec and GauSpec, the estimator of $\boldsymbol{X}$ is given by $\hat{\boldsymbol{x}}_{\text{GS}} = \hat{\boldsymbol{x}}_{\text{OS}} = \hat{\boldsymbol{x}}_{\text{Spec}} \coloneqq \sqrt{N}\,\overline{\boldsymbol{v}}_N$. Notice that, while OptSpec requires knowing exactly the statistics of the noise and the SNR, GauSpec

represents the spectral estimator that would be used by a statistician who assumes the noise to be Gaussian and the SNR $\lambda$ (instead of $\lambda_*$). An application of Theorem 2.2 in [22] gives that

$$\mathrm{MSE}(M_{\mathrm{OS}}) = \frac{1}{2}\big(1 - C^2(\rho, \lambda_*)\big), \quad \text{and} \quad \mathrm{MSE}(M_{\mathrm{GS}}) = \frac{1}{2}\big(1 + (1 - 1/\lambda)^2 - 2(1 - 1/\lambda)C(\rho, \lambda_*)\big).$$

Furthermore, the respective overlaps coincide and are given by $\sqrt{C_{\mathrm{OS}}}$. Note that $\mathrm{MSE}(M_{\mathrm{GS}}) = \mathrm{MSE}(M_{\mathrm{OS}}) + \frac{1}{2}(C(\rho, \lambda_*) - (1 - 1/\lambda))^2$. Thus, as expected, we have $\mathrm{MSE}(M_{\mathrm{OS}}) \le \mathrm{MSE}(M_{\mathrm{GS}})$. We also remark that, as the noise matrix is rotationally invariant, both the MSE and the overlap of the spectral estimators do not depend on the ground-truth signal distribution.

## 3 Main results: performance of inference algorithms

### 3.1 Mismatched Bayes estimator

Consider the following functions of $\lambda, \lambda_* > 0$ and of the asymptotic spectral noise density $\rho$:

$$
\begin{aligned}
M(\lambda, \lambda_*) &:= \Big(1 - \frac{1}{\sqrt{\lambda\lambda_*}}\Big)\Big(1 - \frac{1}{\lambda_*}R_\rho'\Big(\frac{1}{\sqrt{\lambda_*}}\Big)\Big)\mathbf{1}\big(\overline{h}\sqrt{\lambda_*} \ge 1 \cap \lambda\lambda_* > 1\big), \\
Q(\lambda, \lambda_*) &:= \Big(1 - \frac{1}{\sqrt{\lambda\lambda_*}}\Big)^2\mathbf{1}\big(\overline{h}\sqrt{\lambda_*} \ge 1 \cap \lambda\lambda_* > 1\big) + \Big(1 - \frac{\overline{h}}{\sqrt{\lambda}}\Big)^2\mathbf{1}\big(\overline{h}\sqrt{\lambda_*} < 1 \cap \sqrt{\lambda} > \overline{h}\big),
\end{aligned}
\tag{9}
$$

where $\mathbf{1}(E)$ stands for the indicator function of the event $E$. The idea is that $Q(\lambda, \lambda_*)$ and $M(\lambda, \lambda_*)$ represent the squared norm of the mismatched Bayes estimator and its inner product with the signal, respectively. This leads to our main result on the mismatched Bayes MSE, which is stated below.

**Theorem 1** (Performance of mismatched Bayes estimator). *Consider a spiked model* (1). *Let Assumptions 1 and 2 hold. Then, the MSE of the mismatched Bayes estimator* (3) *is given by*

$$\mathrm{MSE}(M_{\mathrm{mis}}) = \frac{1}{2}\big(1 + Q(\lambda, \lambda_*) - 2M(\lambda, \lambda_*)\big). \tag{10}$$

As a consequence of the auxiliary lemmas used in the proof of Theorem 1 and presented in Appendix B, we have that $N^{-2}\int P_{\mathrm{mis}}(d\boldsymbol{x}\,|\,\boldsymbol{Y})\langle\boldsymbol{X},\boldsymbol{x}\rangle^2$ converges almost surely to $M(\lambda, \lambda_*)$. Furthermore, for noise matrices regularized by arbitrarily small Wigner matrices, by using similar techniques as in [15] we can prove that $\|M_{\mathrm{mis}}\|_{\mathrm{F}}/N$ converges almost surely to $Q^{1/2}(\lambda, \lambda_*)$. We conjecture that this convergence holds even without the Wigner regularization. By assuming this conjecture, we have the following almost sure convergence result for the overlap:

$$\mathrm{Overlap}_{\mathrm{mis}} = \frac{M^{1/2}(\lambda, \lambda_*)}{Q^{1/4}(\lambda, \lambda_*)}, \tag{11}$$

which holds whenever the denominator is non-zero. When comparing the overlaps of different methods in Section 4, we will use (11) for the mismatched Bayes estimator.

Let us highlight the following property of the MSE (10) when there is no mismatch in the SNR:

$$\text{If } \lambda = \lambda_* \text{ and } \overline{h} \ge 1, \text{ then } \mathrm{MSE}(M_{\mathrm{mis}}) = \mathrm{MSE}(M_{\mathrm{GS}}). \tag{12}$$

However, in general $\mathrm{MSE}(M_{\mathrm{mis}})$ and $\mathrm{MSE}(M_{\mathrm{GS}})$ differ. E.g., when the assumed SNR $\lambda$ is different from the real one $\lambda_*$, the norm of $M_{\mathrm{mis}}$ incorporates information about $\lambda_*$ while the one for $M_{\mathrm{GS}}$ only depends on $\lambda$. Finally, we note that, from a statistical view-point, Assumption 1 can be replaced by the following: $\boldsymbol{X}$ is uniformly distributed on $\mathbb{S}^{N-1}(\sqrt{N})$ and is independent of $\boldsymbol{Z}$, which may be a generic symmetric matrix with converging empirical spectral density.

**Sketch of the proof of Theorem 1.** The goal is to evaluate the asymptotic values of $\mathbb{E}\|M_{\mathrm{mis}}\|_{\mathrm{F}}^2$ and $\mathrm{tr}\,\mathbb{E}[M_{\mathrm{mis}}\boldsymbol{X}\boldsymbol{X}^\top]$, from which both the MSE and the mean overlap of the Bayes estimator can be obtained. Our strategy is to first compute the log-moment generating function $f_{\mathrm{mis}}$ of the mismatched Bayes model. Then, the above quantities of interest can be accessed by taking derivatives of $f_{\mathrm{mis}}$ with respect to appropriate parameters. However, as the noise in the inference model is not Gaussian, $\mathbb{E}\|M_{\mathrm{mis}}\|_{\mathrm{F}}^2$ cannot be computed in the standard way using an "I-MMSE" type of formula [46] (like it is done, e.g., in [71]). To address this issue, we introduce a generalized model for which the noise matrix is given by the original one plus an independent Wigner matrix multiplied by a small parameter $\epsilon$. We then compute the log-moment generating function of this generalized model using the theory of low-rank perturbations of rotationally invariant random matrices [21] and of the low-rank spherical integral [45]. Finally, the desired results are obtained by a limiting argument in $\epsilon \to 0$.

## 3.2 Approximate message passing

Our main result is a characterization of the iterates (8) in the high-dimensional limit. We show that the joint empirical distribution of $(\boldsymbol{x}^1, \ldots, \boldsymbol{x}^t, \hat{\boldsymbol{x}}^1, \ldots, \hat{\boldsymbol{x}}^{t+1})$ converges (in $W_2$) to the random vector $(x_1, \ldots, x_t, \hat{x}_1, \ldots, \hat{x}_{t+1})$. The law of the random vector $(x_1, \ldots, x_t, \hat{x}_1, \ldots, \hat{x}_{t+1})$ is captured by a deterministic *state evolution (SE)* recursion, which can be expressed via a sequence of vectors $\boldsymbol{\mu}_t = (\mu_1, \ldots, \mu_t)$ and matrices $\boldsymbol{\Sigma}_t, \boldsymbol{\Delta}_t, \boldsymbol{B}_t \in \mathbb{R}^{t \times t}$ defined as follows. We start with the initialization

$$\mu_1 = \sqrt{\lambda_*}\epsilon, \quad \boldsymbol{\Sigma}_1 = \bar{\kappa}_2 \mathbb{E}[\hat{x}_1^2], \quad \boldsymbol{\Delta}_1 = \mathbb{E}[\hat{x}_1^2], \quad \boldsymbol{B}_1 = \bar{\kappa}_1, \tag{13}$$

where $\lambda_*$ is the true SNR (see (1)), and $\{\bar{\kappa}_k\}_{k \geq 1}$ are the free cumulants associated to the asymptotic spectral measure of the noise $\rho$. For $t \geq 1$, given $\boldsymbol{\mu}_t, \boldsymbol{\Sigma}_t, \boldsymbol{\Delta}_t, \boldsymbol{B}_t$, let

$$\begin{cases} (z_1, \ldots, z_t) \sim \mathcal{N}(\boldsymbol{0}, \boldsymbol{\Sigma}_t) \text{ and independent of } (X, \hat{x}_1), \\ x_t = z_t + \mu_t X - \bar{\beta}_t \hat{x}_{t-1} + \sum_{i=1}^t (\boldsymbol{B}_t)_{t,i} \hat{x}_i, \quad \hat{x}_{t+1} = h_{t+1}(x_t), \end{cases} \tag{14}$$

where we have defined $\hat{x}_0 = 0$, $\bar{\beta}_1 = 0$, and for $t \geq 2$, $\bar{\beta}_t = \mathbb{E}[h_t'(x_{t-1})]$. Then, the parameter $\mu_{t+1}$ is

$$\mu_{t+1} = \sqrt{\lambda_*} \mathbb{E}[X \hat{x}_{t+1}]. \tag{15}$$

Next, we compute the matrices $\boldsymbol{\Delta}_{t+1}$ and $\boldsymbol{B}_{t+1}$ as

$$\begin{cases} (\boldsymbol{\Delta}_{t+1})_{i,j} = \mathbb{E}[\hat{x}_i \hat{x}_j], \quad 1 \leq i, j \leq t+1, \\ \boldsymbol{B}_{t+1} = \sum_{j=0}^t \bar{\kappa}_{j+1} \bar{\boldsymbol{\Phi}}_{t+1}^j \text{ where } (\bar{\boldsymbol{\Phi}}_{t+1})_{i,j} = 0 \text{ if } i \leq j, \text{ and } (\bar{\boldsymbol{\Phi}}_{t+1})_{i,j} = \mathbb{E}[\partial_j \hat{x}_i] \text{ if } i > j, \end{cases} \tag{16}$$

where $\partial_j \hat{x}_i$ denotes the partial derivative $\partial_{z_j} h_i(z_{i-1} + \mu_{i-1} X - \bar{\beta}_{i-1} \hat{x}_{i-2} + \sum_{k=1}^{i-1} (\boldsymbol{B}_t)_{i-1,k} \hat{x}_k)$. Finally, we define the covariance matrix $\boldsymbol{\Sigma}_{t+1}$ as

$$\boldsymbol{\Sigma}_{t+1} = \sum_{j=0}^{2t} \bar{\kappa}_{j+2} \boldsymbol{\Theta}_{t+1}^{(j)}, \quad \text{with} \quad \boldsymbol{\Theta}_{t+1}^{(j)} = \sum_{i=0}^j (\bar{\boldsymbol{\Phi}}_{t+1})^i \boldsymbol{\Delta}_{t+1} (\bar{\boldsymbol{\Phi}}_{t+1}^\top)^{j-i}. \tag{17}$$

Note that the $t \times t$ top left sub-matrices of $\boldsymbol{\Sigma}_{t+1}, \boldsymbol{\Delta}_{t+1}$ and $\boldsymbol{B}_{t+1}$ are given by $\boldsymbol{\Sigma}_t, \boldsymbol{\Delta}_t$ and $\boldsymbol{B}_t$. At this point, we are ready to state our state evolution characterization of the AMP algorithm (8). The result is presented in terms of *pseudo-Lipschitz* test functions. A function $\psi : \mathbb{R}^m \to \mathbb{R}$ is pseudo-Lipschitz of order 2, i.e., $\psi \in \mathrm{PL}(2)$, if there is a constant $C > 0$ such that $\|\psi(\boldsymbol{x}) - \psi(\boldsymbol{y})\| \leq C(1 + \|\boldsymbol{x}\| + \|\boldsymbol{y}\|)\|\boldsymbol{x} - \boldsymbol{y}\|$. We also make the following assumption on the functions $\{h_{t+1}\}_{t \geq 1}$.

**Assumption 3.** *The function $h_{t+1} : \mathbb{R} \to \mathbb{R}$ is Lipschitz, and the partial derivatives*

$$\partial_{z_k} h_{t+1}\Big(z_t + \mu_t X - \bar{\beta}_t \hat{x}_{t-1} + \sum_{k=1}^t (\boldsymbol{B}_t)_{t,k} \hat{x}_k\Big)$$

*are continuous on a set of probability 1, under the laws of $(z_1 \ldots, z_t)$ and $(\hat{x}_1, \ldots, \hat{x}_t)$ given in (14).*

This requirement covers most practically relevant choices of $h_{t+1}$ (e.g., soft-thresholding or ReLU), and it is rather standard in the AMP literature, see e.g. [39, 60, 82, 87].

**Theorem 2** (State evolution of Gaussian AMP)**.** *Consider a spiked model* (1) *and the AMP algorithm* (8) *initialized as in* (7)*. Let Assumptions 1 and 3 hold, and let $\psi : \mathbb{R}^{2t+2} \to \mathbb{R}$ be any pseudo-Lipschitz functions of order 2. Then, for each $t \geq 1$, we almost surely have that, as $N \to +\infty$,*

$$\frac{1}{N} \sum_{i=1}^N \psi((\boldsymbol{x}^1)_i, \ldots, (\boldsymbol{x}^t)_i, (\hat{\boldsymbol{x}}^1)_i, \ldots, (\hat{\boldsymbol{x}}^{t+1})_i, (\boldsymbol{X})_i) \to \mathbb{E}\psi(x_1, \ldots, x_t, \hat{x}_1, \ldots, \hat{x}_{t+1}, X), \tag{18}$$

*where the random variables on the right are defined in* (14)*.*

By using Definition 6.7 and Theorem 6.8 of [84], one readily obtains that (18) is equivalent to the convergence of $(\boldsymbol{x}^1, \ldots, \boldsymbol{x}^t, \hat{\boldsymbol{x}}^1, \ldots, \hat{\boldsymbol{x}}^{t+1}, \boldsymbol{X})$ to $(x_1, \ldots, x_t, \hat{x}_1, \ldots, \hat{x}_{t+1}, X)$ in $W_2$ distance. Applying (18) to the pseudo-Lipschitz functions $\psi(\hat{x}_{t+1}, X) = (\hat{x}_{t+1} - X)^2$, $\psi(\hat{x}_{t+1}, X) = \hat{x}_{t+1} \cdot X$ and $\psi(\hat{x}_{t+1}, X) = (\hat{x}_{t+1})^2$, we obtain a high-dimensional characterization of the AMP performance.

**Corollary 1.** *Consider the setting of Theorem 2. Then, for each $t \geq 1$, we almost surely have that*

$$\mathrm{MSE}(M_{\mathrm{AMP}}^t) := \lim_{N \to +\infty} \frac{1}{2N^2} \mathbb{E}\|\boldsymbol{X}\boldsymbol{X}^\top - \hat{\boldsymbol{x}}^t(\hat{\boldsymbol{x}}^t)^\top\|_{\mathrm{F}}^2 = \frac{1}{2}\Big(1 - 2\left(\mathbb{E}[\hat{x}_t \cdot X]\right)^2 + \left(\mathbb{E}[(\hat{x}_t)^2]\right)^2\Big),$$

$$\mathrm{Overlap}(\hat{\boldsymbol{x}}^t) := \lim_{N \to +\infty} \frac{|\langle \boldsymbol{X}, \hat{\boldsymbol{x}}^t \rangle|}{\|\hat{\boldsymbol{x}}^t\| \cdot \|\boldsymbol{X}\|} = \frac{|\mathbb{E}[\hat{x}_t \cdot X]|}{\sqrt{\mathbb{E}[(\hat{x}_t)^2]}}.$$

**Sketch of the proof of Theorem 2.** The key insight is to exploit the flexibility given by the denoisers of the abstract AMP iteration [87] to "correct" the mismatch. These denoisers are denoted by $\{\tilde{h}_t\}_{t\geq 2}$ (see (33) and (34) in Appendix C.1) and they serve solely as a proof technique, hence they do not have an impact on the practicality of the algorithm; in contrast, the denoisers $\{h_t\}_{t\geq 2}$ (see (8)) can be chosen by the algorithm designer to take advantage of the information available about the signal. Thus, we can design an *auxiliary AMP* such that *(i)* it admits a state evolution characterization, and *(ii)* its iterates are close to (8). This requires the delicate induction argument in Appendix C: in Appendix C.1, we define the auxiliary AMP; in Appendix C.2, we prove a state evolution result for it; in Appendix C.3, we show by induction that *(i)* the state evolution parameters of the auxiliary AMP coincide with those defined in this section, and *(ii)* the iterates of the auxiliary AMP are close (in $\ell_2$ norm) to the iterates of the AMP (8), thus concluding the proof of Theorem 2.

## 4   Numerical results and discussion

In all experiments, the density $\rho$ of the eigenvalues of the noise has unit variance and the distribution of the ground-truth signal is uniform on the sphere of radius $\sqrt{N}$. For the sake of simplicity, we decouple the effect of the mismatch in *(i)* the noise statistics and *(ii)* the SNR. More specifically, in the first two examples, we set $\lambda = \lambda_*$, so that only mismatch in the noise statistics is present; and in the last example, the noise is Gaussian, so that there is only SNR mismatch. Additional results when $\boldsymbol{Z}$ is the free convolution of Rademacher and semicircle spectra are reported in Appendix D.3.

**Models of mismatch.** *(a) Marcenko-Pastur spectrum.* An example of non-symmetric density is the Marcenko-Pastur law $\rho(x) = \sqrt{x(4-x)}/(2\pi x)\, \mathbf{1}(x \in [0,4])$. The $R$-transform is $R_\rho(x) = 1/(1-x)$, and the results are displayed in Fig. 1a (the same formulas apply if the law is centered).

*(b) Uniform spectrum.* We let $\rho$ be the uniform distribution $\mathcal{U}[-\sqrt{3}, \sqrt{3}]$. In this case, we have $R_\rho(x) = \sqrt{3}/\tanh(\sqrt{3}x) - 1/x$, and the results are in Fig. 1b.

*(c) Wigner matrix/semicircle spectrum with mismatched SNR.* We consider a Gaussian noise matrix $\boldsymbol{W}$ (as assumed by the statistician), but with mismatched SNR by setting $\lambda = 4\lambda_*$, see Fig. 1c.

We remark that Assumption 1 is verified for all models. Furthermore, in all the three cases, the support of $\rho$ is a single non-empty interval. Hence, Theorem 2.2 of [10] and Proposition 2.4 of [21] can be used to show that Assumption 2 holds.

**Inference algorithms and set-up.** The estimators of the spike $\boldsymbol{X}\boldsymbol{X}^\top$ are compared in terms of the MSE (4) (plots on the left). The AMP and spectral estimators of $\boldsymbol{X}$ are compared in terms of the overlap (5), and the Bayes estimator of $\boldsymbol{X}$ in terms of the overlap (6) (plots on the right). We evaluate those formulas at $N = 8000$ for the various algorithms, and in the $N \to +\infty$ limit for the theoretical predictions. The *correct AMP* (together with its own SE) is in red. This AMP is *correct* in the sense that the statistician is aware of the noise statistics and, thus, incorporates the right Onsager corrections. The form of the correct AMP and the corresponding state evolution are readily obtained from the results in [37, 87] and, for the reader's convenience, they are recalled in Appendix D.1. As non-linearities, we use the posterior-mean denoising functions of Section 5.1 in [87], and we estimate the SE parameters consistently from data (see Appendix D.2). The *Gaussian AMP* (together with its own SE) is in blue. This is the AMP algorithm (8), which is chosen when the noise statistics or the SNR are unknown (as it would be optimal for Gaussian noise and known SNR). Its state evolution is given by Theorem 2. As non-linearities $\{h_{t+1}\}_{t\geq 1}$, we use again the posterior-mean denoising functions, and estimate consistently the state evolution parameters from data (see Appendix D.2). We note that the denoisers of the Gaussian AMP depend only on a single iterate, while the denoisers of the correct AMP incorporate all the past iterates. In contrast, the state evolution parameters of the Gaussian AMP at time $t$ depend on *all* the past, in order to correct for the mismatch. The *mismatched Bayes estimator* is plotted in green. Its MSE and overlap are given by (10) and (11), respectively. The *optimal spectral estimator (OptSpec)* is plotted in black, and the *Gaussian mismatched spectral estimator (GauSpec)* is plotted in yellow in Fig. 1a and 1c (where it differs from Bayes) or in green in Fig. 1b (where it coincides with Bayes). The performance measures of both variants of spectral estimators are given at the end of Section 2.3. Each experiment is repeated for 10 independent runs. We report the average and error bars at 1 standard deviation.

**A rich and surprising phenomenology.** *(i)* An intriguing phenomenon is that, in terms of MSE, *the Gaussian AMP does not perform as well as the mismatched Bayes estimator*. This effect occurs

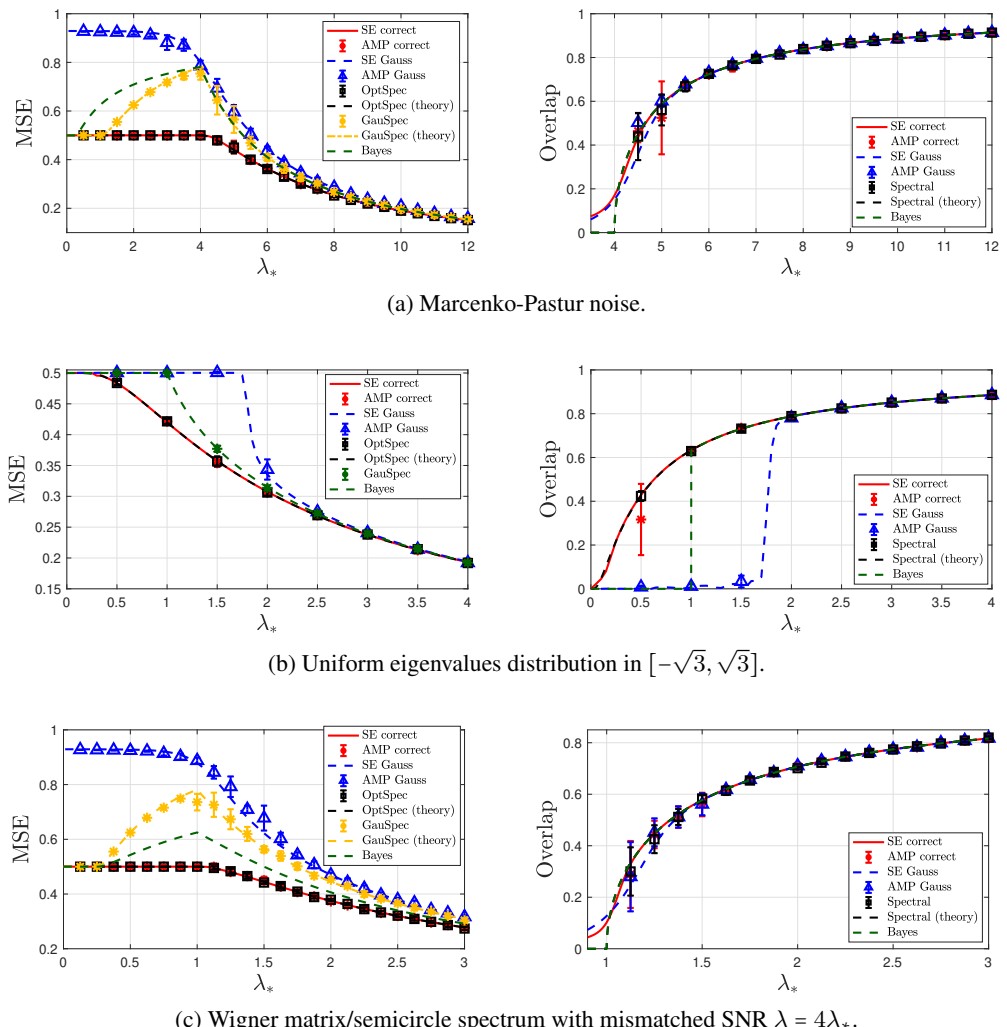

(a) Marcenko-Pastur noise.

(b) Uniform eigenvalues distribution in $[-\sqrt{3}, \sqrt{3}]$.

(c) Wigner matrix/semicircle spectrum with mismatched SNR $\lambda = 4\lambda_*$.

Figure 1: MSE (on the left) and overlap (on the right), as a function of the true SNR $\lambda_*$ in three mismatched settings.

in all our settings, and it is surprising as the Gaussian AMP (8) is precisely designed to efficiently compute the Bayes estimator (3). In the Bayes-optimal case, it does so (outside of its computational gap) [12]. However, if there is a mismatch, *AMP does not compute the (mismatched) posterior mean*. This finding adds to the already existing evidence [5] that AMP is poorly understood, and studying the fundamental reasons behind this behavior is an exciting avenue for future research.

*(ii)* By comparing MSE[2] and overlap curves, we understand that the discrepancy between Gaussian AMP and Bayes comes from an incorrect estimation of the signal norm. In fact, *at large enough SNR, the overlaps of Bayes and Gaussian AMP match, and they even coincide with the overlaps of optimal algorithms* (correct AMP and OptSpec), which exploit knowledge about the noise structure. Understanding the origin of the MSE discrepancy (namely, a wrong estimate of the signal length) can potentially lead to procedures which correct for this effect and, hence, reduce the MSE.

*(iii)* When there is no SNR mismatch and $\bar{h} \geq 1$ (cf. (12)), *the MSEs of the Bayes and Gaussian spectral estimators match* (see Fig. 1b). Both have no information about the noise structure and about the signal distribution. Yet, this equality is remarkable given that the spectral estimator is the solution of an optimization problem (it is a "zero temperature" estimator in physics parlance), while the Bayes

---

[2]The MSE is constructed from the norm $\mathbb{E}\|M(\boldsymbol{Y})\|_{\mathrm{F}}^2$ and the inner product $\mathrm{tr}\mathbb{E}[M(\boldsymbol{Y})\boldsymbol{X}\boldsymbol{X}^{\top}]$.

estimator aims at computing the mean of a certain posterior distribution (it is a "finite temperature" estimator).

*(iv) All mismatched estimators are outperformed by the optimal spectral method and the correct AMP.* Both these algorithms take advantage of the noise statistics and achieve the same MSE. This *suggests that the two estimators are Bayes-optimal*. We remark that solving this conjecture would require understanding the information-theoretic limits of low-rank estimation with structured noise.

*(v)* Finally, a striking phenomenon – first observed in [71] – is that *the Bayes and Gaussian spectral MSE curves may be non-decreasing with the true SNR $\lambda_*$*, see Figs. 1a and 1c. Going beyond the analysis in [71], we remark that, for large enough SNR, all estimators yield the same overlap and therefore all "point in the correct direction" given by the leading eigenvector, see also part *(ii)* of this discussion. This links the initial MSE increase to a wrong estimation of the signal's norm, due to an over-confidence in the data quality (recall that the assumed SNR $\lambda$ is equal to $4\lambda_*$ in Fig. 1c).

## Acknowledgements

M. Mondelli was partially supported by the 2019 Lopez-Loreta Prize. The authors acknowledge discussions with A. Krajenbrink, M. Robinson, A. Depope, N. Macris and F. Pourkamali.

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
