# OpenReview forum: "The price of ignorance: how much does it cost to forget noise structure in low-rank matrix estimation?"
_NeurIPS.cc/2022/Conference — NeurIPS 2022 Accept_

### Official Review · Reviewer_oMyw · 2022-06-26

**Rating:** 6
**Confidence:** 4
**Soundness:** 4 excellent
**Presentation:** 4 excellent
**Contribution:** 2 fair

**Summary:**

This paper presents a theoretical analysis of the performance of two algorithms for rank-1 matrix recovery under mismatched noise. Specifically, a closed-form MSE for the Bayes estimator and a state evolution analysis of the AMP algorithm are provided. The paper also discusses some phenomena arising from numerical experiments regarding these algorithms under two different performance metrics.

**Questions:**

The discussion about Bayes estimator and AMP from the numerical results is interesting, but it might be helpful to provide more insights into the performance gap of AMP and Bayes in this problem setup, in addition to this phenomenon itself.

I understand that this is a theoretical work and if the authors think these are out of scope for this work, but I'd appreciate it if the authors can provide some comments about the applicability of the algorithms discussed in this paper. Under what type of scenarios might one favor one approach over the other?


**Limitations:**

While this is very theoretical work that proves performance guarantees, a dedicated discussion of the limitations of the scope and applicability of the results might still be helpful.

**Strengths And Weaknesses:**

Strengths:
- This paper gives theoretically solid performance guarantees for the Bayes estimator and AMP algorithm for mismatched rank-1 matrix recovery. Settings with unknown noise distributions are common in real-world applications, where this work helps to understand the effect of these algorithms under mismatched noise.
- Numerical experiments comparing the theoretical, experimental, and noise-aware performances are presented, which corroborate the results of this paper and shed light on the points of discussion.
- The writing and presentation of this paper is clear.

Weaknesses:
- Aside from the technical contributions, I am not fully convinced of the level of impact that this paper makes. This paper analyzes two existing algorithms with mismatched spherically symmetric noise, which is a step up from the analysis in [71] of mismatched Gaussian SNR, but might not suffice for this venue. Would consider raising the score if this concern is properly addressed.

---

> ### Author Response · Authors · 2022-08-01
> **Response to Reviewer oMyw (part 1/2)**
>
> We thank the reviewer for the useful comments. We reply to the various points raised in the review below, and we are happy to answer additional follow-up questions. In case the reviewer is satisfied with our responses (especially the first one concerning the impact of our contribution), we hope that raising the rating will be considered.
>
> ----
>
> *"I am not fully convinced of the level of impact that this paper makes. This paper analyzes two existing algorithms with mismatched spherically symmetric noise, which is a step up from the analysis in [71] of mismatched Gaussian SNR"*
>
> We clarify below the impact of our contribution which goes well beyond the existing analysis in [71].
>
> First, let us highlight that the assumptions of our paper are much weaker than assuming a ‘mismatched spherically symmetric noise’. In fact, our only requirement on the noise is that it is rotationally invariant, namely the eigenvector matrices are uniformly sampled among rotation matrices. This allows for an arbitrary spectrum of the noise, and it can therefore model the complex correlations that often appear in application domains. Indeed, only in the case of the semicircle law for the noise eigenvalues, the entries of the noise are independent. For any other choice, the matrix entries are dependent, and this is covered by our analysis. This is not the case in [71]. We emphasize that taking into account dependent random variables in such precise high-dimensional analyses is usually considered a rather involved task and a strong generalization, and has been much less studied than gaussian/independent settings for that reason (as we point out in the introduction).
>
> Second, while [71] simply proposes a proof strategy, the results in our manuscript are fully rigorous. In fact, following the strategy suggested in [71] might prove rather difficult, since it would require a very precise control of many error terms when applying Laplace’s method. Having a completely rigorous analysis is key in order to assess the unexpected phenomenology we discuss in the experimental part of the paper. In particular, given the interest of a broad community concerning AMP algorithms, we believe that our surprising discoveries on its behavior will trigger new research directions.
>
> Finally, in [71] the authors mention as an interesting open problem the performance comparison between the Bayes estimator and AMP. This is exactly what our contribution is able to achieve. In fact, we provide rigorous performance guarantees on both a mismatched Bayes estimator (Theorem 1) and an AMP algorithm (Theorem 2), all this in a much more general setting than the one studied in [71]. Let us conclude by mentioning that our paper not only provides analytical results, but it also uses such results to display the remarkable phenomenology presented in Section 4. This phenomenology is absent in [71], and we regard it among the most surprising results of our contribution.
>
> Given all this and the fact that rank-one matrix estimation is one of the most paradigmatic models of high-dimensional inference, we foresee an impact in the broad and interdisciplinary community of researchers working on the fundamental and algorithmic limits of inference and message-passing algorithms.
>
> ----
>
> *"The discussion about Bayes estimator and AMP from the numerical results is interesting, but it might be helpful to provide more insights into the performance gap of AMP and Bayes in this problem setup"*
>
> This is an excellent point. Our numerical simulations show that the overlap of Gaussian AMP and of the Bayes estimator match at high SNR, see the plots on the right in Fig. 1. Hence, this performance gap must be due to an incorrect estimation of the norm of the signal, see also the discussion at l. 347-352 of the revision. It is an open problem (and an exciting direction for future research) to bridge the gap between the Gaussian AMP algorithm and the mismatched Bayes estimator. See also the answer 3 to reviewer *RAkD*.

---

> ### Author Response · Authors · 2022-08-01
> **Response to Reviewer oMyw (part 2/2)**
>
> *"I'd appreciate it if the authors can provide some comments about the applicability of the algorithms discussed in this paper. Under what type of scenarios might one favor one approach over the other?"*
>
> In this paper, we study the effect of mismatch on three types of inference procedures: a Bayes estimator, AMP algorithms, and spectral methods. Spectral methods consist in suitably rescaling the principal eigenvector of the data matrix. The scaling factor takes into account the spectrum of the data matrix and it is chosen in order to minimize the MSE. This approach is unable to take advantage of any information on the signal prior. In fact, as the noise matrix is rotationally invariant, both the MSE and the overlap of the spectral estimator do not depend on the signal prior (see l. 225-226 of the revision). In contrast, in the AMP algorithm (8), the algorithm designer can choose the denoiser $h_t$ in order to adapt to the signal prior (see l. 199-200 of the revision). This is a key advantage of AMP and it is what makes it Bayes-optimal in certain (matched) settings, see e.g. [29, 30, 61, 13]. Finally, the Bayes estimator directly samples from the posterior, but it does not lead to a practical algorithm (see l. 176 in the revision).
>
> ----
>
> *"a dedicated discussion of the limitations of the scope and applicability of the results might still be helpful"*
>
> This is a very good point. As mentioned by the reviewer, our contribution is of theoretical nature. Thus, an in-depth investigation of the practical applicability of our findings is out of the scope of our paper. At the same time, AMP algorithms have been recently considered for several practical scenarios. Examples include problems related to genetics (see [86]), inpainting (see [66]) and MRI image recovery (see [76]). Thus, we can foresee an application of the results of this paper in such domains of applications. We have included reference [76] in the revision and edited in l. 37-38. Furthermore, if the reviewer thinks it would be helpful, we could incorporate a brief discussion on the scope and applicability of our results in a revision.

---

> > ### Comment · Reviewer_oMyw · 2022-08-08
> > **Thanks for the response**
> >
> > Thank you to the authors for the detailed response, to which I am satisfied. I will raise the rating from 5 to 6.

---

### Official Review · Reviewer_RAkD · 2022-07-10

**Rating:** 6
**Confidence:** 3
**Soundness:** 3 good
**Presentation:** 2 fair
**Contribution:** 3 good

**Summary:**

The authors considered the problem of rank-one signal estimation corrupted by structured rotationally invariant noise, and studied the situation when the actual noise statistics deviates from the Gaussian assumption. Three estimation procedures were proposed and discussed: mismatched Bayes estimator, approximate message passing, and spectral estimators. Both theoretical analysis and simulation experiments were provided to compare these different methods, together with reasonable conclusions and interesting insights.

**Questions:**

The authors mentioned that the performance gap between the AMP algorithm and the Bayes estimator is due to an incorrect estimation of the signal norm. A natural question is whether the AMP algorithm could be modified accordingly, so that a better estimation of signal norm can be achieved, so as to narrow down the gap to the Bayes estimator.

**Limitations:**

The paper focused on theoretical analysis and simulated experiments. It would be nice if the authors can relate the research to some real world scenarios, and show some experimental results on real data.

**Strengths And Weaknesses:**

Strengths:
1) A thorough study on the mismatched estimation problem for rank-one matrices was provided, with in-depth theoretical analysis and interesting empirical findings.
2) Different estimation methods were proposed and compared, showing the strengths and weaknesses of each method.
3) An interesting phenomenon was identified that Gaussian AMP does not perform as well as the mismatched Bayes estimator. Discussions were provided on potential causes of this phenomenon.

Weaknesses:
1) The Gausian AMP algorithm was provided in equation (8), but not much details were provided on how it is derived. It is unclear what objective function this procedure is optimizing. More technical details would help the readers to better understand the algorithm.
2) Similarly, the mismatch performance metrics were provided in equation (9), but not much details were provided on the meaning of these metrics. Better explanations of these metrics may help the readers to better understand the motivations behind the equations.

---

> ### Author Response · Authors · 2022-08-01
> **Response to Reviewer RAkD**
>
> We thank the reviewer for the useful feedback. We reply to the various points raised in the review below.
>
> ----
>
> *"The Gaussian AMP algorithm was provided in equation (8), but not much details were provided on how it is derived."*
>
> This is an excellent point. A very good pointer here is the recent review [39], which discusses in detail how such an AMP can be derived. We have updated the paper accordingly (see l. 203-204 of the revision).
>
> ----
>
> *"the mismatch performance metrics were provided in equation (9), but not much details were provided on the meaning of these metrics"*
>
> We thank the reviewer for this remark, which has allowed us to improve the clarity of Theorem 1. In fact, the quantities $M(\lambda,\lambda_*)$ and $Q(\lambda,\lambda_*)$
> have a simple interpretation in terms of the mismatched Bayes estimator: $Q(\lambda,\lambda_*)$ represents the squared norm of the mismatched Bayes estimator and $M(\lambda,\lambda_*)$ represents its inner product with the signal. We have edited l. 230-232 of the revised version to address this comment.
>
> ----
>
> *"A natural question is whether the AMP algorithm could be modified accordingly"*
>
> This is also an excellent point. If the designer is aware of the noise statistics, then one natural choice is what we call the ‘correct AMP’. This algorithm is used for comparison in the numerical results of Section 4, it can be obtained from [37, 87] and the details are provided in Appendix D of the supplementary material. In contrast, if the designer still incorrectly assumes Gaussian statistics, it is an open problem to bridge the gap between the Gaussian AMP algorithm and the mismatched Bayes estimator. Given that our numerical results show that the performance gap is due to an incorrect signal’s norm estimation, we may consider running Gaussian AMP and a-posteriori normalizing its final estimate (like for the spectral estimators). Thanks for the interesting suggestion that we will consider for future work.
>
> ----
>
> *"It would be nice if the authors can relate the research to some real world scenarios"*
>
> Thank you for this remark. We would like to clarify that providing experimental results on real data is out of scope for the current contribution, whose goal is to provide a rigorous performance analysis for various estimators (Bayes, AMP, spectral) in the presence of mismatch. Having said that, we also remark that AMP algorithms have been recently considered for several practical scenarios. Examples include problems related to genetics (see [86]), inpainting (see [66]) and MRI image recovery (see [76]). Thus, we can foresee an application of the results of this paper in such domains of applications. We have included reference [76] in the revision and edited in l. 37-38.

---

> > ### Comment · Reviewer_RAkD · 2022-08-07
> > **Final comments**
> >
> > Thanks the authors for responding to my previous questions.
> > My rating for this submission remains unchanged (6: Weak Accept).

---

### Official Review · Reviewer_ggxP · 2022-07-10

**Rating:** 6
**Confidence:** 3
**Soundness:** 3 good
**Presentation:** 3 good
**Contribution:** 2 fair

**Summary:**

The paper considers the penalty to be paid when recovering a rank-1 matrix from noisy partial observations when the noise is assumed to be Gaussian in the case where the noise isn't so. The paper's analysis considers two algorithms and regimes: a Bayes estimator in an asymptotic regime and an approximate message passing algorithm in a state evolution regime. The analysis finds a gap in the performance of the two estimators that depends on the level of noise due to an inaccurate estimation of the signal norm in AMP.


**Questions:**

For the rescaled overlap measure (5), is it the same as the asymptotic the cosine distance between $\bf X$ and $\bf{\hat{x}}({\bf Y})$? If yes, this may be easier to describe. Additionally, the concern on the Bayes estimator in line 179 seems to be independent of the overlap measure - would it not also apply to the MSE?

Where is the function ht in (8) described? The sketch of the proof of Theorem 2 refers to "choosing denoisers carefully". Can there be a more explicit discussion of what "carefully" choosing denoisers entails in practice?

In line 254, should "converges... to the random vector" be "converges... to that of the random vector"? It may help with clarity to state at this point that the second random vector provide the state evolution for this AMP analysis.

**Limitations:**

The paper could use a more detailed discussion on the applicability of Assumptions 1, 2, and 3 in practical settings.

**Strengths And Weaknesses:**

The experimental results show a good match between the performance predictions of the analytical results and the true performance levels from the numerical simulations, and f the dependence of the performance gap between the algorithms on the SNR and noise model mismatch level.

Some instances of simplified notation do not seem necessary, and can detract from clarity (e.g., dropping the index N in line 171, the lack of description of the terminology used in Assumption 2).

The discussion after Theorem 2 would be more clearly stated as a corollary: given that the statement of Theorem 2 appears to be very broad, the authors could provide a statement that is more relevant to assessing the performance of the estimators using metrics like those shown before.

---

> ### Author Response · Authors · 2022-08-01
> **Response to Reviewer ggxP**
>
> We thank the reviewer for the helpful comments. We reply to the various points raised in the review below.
>
> ---------
>
> *"Some instances of simplified notation do not seem necessary"*
>
> We thank the reviewer for this remark. We have changed Eq. (4) and l. 179-180 according to the suggestion. We realize that the paper is rather dense in the notation and the terminology employed (also because of the strict 9 page limit); hence, if the reviewer has any additional suggestions on how to improve the clarity, we would be happy to revise accordingly.
>
> ---------
>
> *"The discussion after Theorem 2 would be more clearly stated as a corollary"*
>
> This is a really good point, and it was also raised by reviewer *MSna*. In the revision, we have added Corollary 1, which gives the expression for the asymptotic MSE and overlap achieved by the Gaussian AMP iterates.
>
> ---------
>
> *"For the rescaled overlap measure (5), is it the same as the asymptotic cosine distance?"*
>
> The rescaled overlap measure (5) and the asymptotic cosine distance are indeed the same, and a comment has been added to note this (see l. 184-185 of the revision).
>
> ---------
>
> *"concern on the Bayes estimator [...] would not also apply to the MSE?"*
>
> This is in fact the case. However, it only applies to the *vector MSE* and not to the *matrix MSE* studied in the paper. In particular, the comment the reviewer refers to implies that $\frac{1}{N} \mathbb E \|\| \int \boldsymbol x P_{{\rm mis}}(d\boldsymbol x\mid \boldsymbol Y) -\boldsymbol X\|\|^2 = 1$ always (which means that the vector MSE is not a proper error measure for this model). However, the (matrix) MSE we analyze is $\frac{1}{2N^2} \mathbb E \|\| M_{\rm mis}(\boldsymbol Y) - \boldsymbol X \boldsymbol X^T\|\|_F^2$, which is non-trivial.
>
> ---------
>
> *"Where is the function ht in (8) described? [...] Can there be a more explicit discussion of what "carefully" choosing denoisers entails in practice?"*
>
> We believe there is a misunderstanding here, and we would like to clarify these points.
>
> First, the function $h_t$ in (8) is *generic* and the algorithm designer is free to choose it. The idea is that $h_t$ can be picked in order to exploit any information available for the signal prior. The only constraint on $h_t$ is given by Assumption 3, which is however rather mild and satisfied by commonly used functions, such as soft-thresholding and ReLU, see also point 1 in the response to reviewer *MSna* and l. 278-279 in the revision.
>
> Second, in the sketch of the proof of Theorem 2, we are referring to the denoisers $\tilde{h}_t$ of the abstract AMP iteration, see (33)-(34) in Appendix C.1. The denoisers $\tilde{h}_t$ need to be chosen carefully, in order to ensure that the AMP iteration (8) (i.e., the object we are interested in analyzing) is close to the AMP iteration (33) (i.e., the object for which a state evolution result can be proved).
>
> Finally, let us highlight that the denoisers that matter in practice are $\{h_t\}$, and the algorithm designer is free to choose them. The denoisers $\{\tilde{h}_t\}$ are introduced solely as a proof technique, in order to show the state evolution result of Theorem 2; hence, they do not have an impact on the practicality of the algorithm.
>
> We hope our explanation answers the question of the reviewer and we have added clarifications at l. 199-200 and 290-292 of the revision.
>
> ---------
>
> *"In line 254, should "converges... to the random vector" be "converges... to that of the random vector"?"*
>
> For $i= 1, \ldots, t$, the empirical distribution of the random vector $\boldsymbol x^i$ converges to the random variable $x_i$. Similarly, for $i= 1, \ldots, t+1$, the empirical distribution of the random vector $\hat{\boldsymbol x}^i$ converges to the random variable $\hat{x}_i$. Thus, the joint empirical distribution in l. 262 “converges… to the random vector”, and not "converges... to that of the random vector".
>
> We agree with the reviewer that it is worth mentioning that the second random vector provides the state evolution, and we have modified the manuscript accordingly (see l. 263 of the revision).
>
> ---------
>
> *"more detailed discussion on the applicability of Assumptions 1, 2, and 3"*
>
> This is a very good point. Some remarks over the generality of Assumptions 1-2 have been added to the text (see l. 139-140 and 151-152 of the revision). Assumption 3 is rather mild and standard in the AMP literature: we have added l. 278-279 in the revision to discuss this point, see also point 1 in the response to reviewer *MSna*.

---

> > ### Comment · Reviewer_ggxP · 2022-08-07
> > **Satisfied with response**
> >
> > Thank you for your response, and particularly for the clarification regarding the two types of denoisers involved in the algorithm analysis. I would suggest to specifically refer to these functions when mentioned using the notation $h_t$ and $\tilde{h}_t$ (e.g., line 290).
> >
> > My rating for this submission remains unchanged.

---

> > > ### Author Response · Authors · 2022-08-08
> > > **Thank you for the positive feedback**
> > >
> > > Thank you for the positive feedback and evaluation of our paper.
> > >
> > > We agree with this last comment, and we have just updated the revision accordingly: we now use the notation $h_t$ and $\tilde h_t$ to refer to the two types of denoisers, see l. 290-294.

---

### Official Review · Reviewer_MSna · 2022-07-17

**Rating:** 6
**Confidence:** 4
**Soundness:** 3 good
**Presentation:** 3 good
**Contribution:** 3 good

**Summary:**

This paper studies rank-1 matrix estimation and provides performance guarantees for two inference methods, a Bayes estimator and an AMP algorithm.

**Questions:**

See "Strengths And Weaknesses".

**Limitations:**

See "Strengths And Weaknesses".

**Strengths And Weaknesses:**

1. The author should comment on whether Assumption 3 holds for some representative examples like soft-thresholding function under the setting considered in this paper.
2. The results and quantities mentioned below Theorem 2 should be formalized into a theorem or corollary.
3. How crucial does Theorem 1 depend on the prior distribution on the signal X (line 163)? Given that the topic of this paper is to study whether these estimators are robust to wrong noise distributional assumption, it doesn't make sense if the result relies highly on the distributional assumption on the signal X (which seems to be an even stronger assumption that might not be correct).
4. When analyzing AMP, the authors assume access to an initialization that is independent of the noise and satisfy certain probabilistic condition (line 186). The authors also claim that in prior literature it is possible to analyze AMP with a practical spectral initialization (line 190). Given the assumed model is already simple enough (rank-1 spiked model), I recommend the authors to improve their analysis to adapt to such a practical initialization scheme.
5. I suggest the authors to make effort discussing their theoretical result in more detail and show how they are related to the main topic of the paper. Currently it is difficult to see this from Theorem 1 and 2.
6. If I am not missing something, the main result for spectral estimators is missing, although from Section 2.3 it seems that there should be some theory for spectral estimators.

---

> ### Author Response · Authors · 2022-08-01
> **Response to Reviewer MSna**
>
> We thank the reviewer for the various detailed comments. We reply to the 6 questions raised in the review below, and we are happy to answer additional follow-up questions. We have also modified both the paper and the appendix accordingly. In case the reviewer is satisfied with our response, we hope that raising the rating will be considered.
>
> ----------------------
>
> **Question 1**
>
> Assumption 3 is rather mild, and it allows us to cover most practically relevant choices of $h_{t+1}$, such as soft-thresholding or ReLU. In fact, $h_{t+1}$ is even allowed to have discontinuous partial derivatives, as long as this discontinuity occurs on a set with 0 measure (this is the case e.g. of ReLU). Let us also point out that such an assumption is standard in the AMP literature: it appears for example in [39] (see Assumption (M2) in Section 3.1), [60] (see the paragraph ‘Continuous differentiability and other technical assumptions’ in Section 3.1), [82] (see Assumption (A1)), and [87] (see Assumption 2.2(b)), among other related works. We have incorporated this comment in l. 278-279 of the revision.
>
> ----------------------
>
> **Question 2**
>
> This is an excellent suggestion, and it was also pointed out by reviewer *ggxP*. In the revision, we have added Corollary 1, which gives the expressions for the asymptotic MSE and overlap achieved by the Gaussian AMP iterates.
>
> ----------------------
>
> **Question 3**
>
> The key point here is that the statistician *assumes* the signal to be spherically distributed. As we analyze a mismatched model, this does not imply that the distribution of the signal *is really* uniform on the sphere. As stated in Assumption 1, we only need that the signal has a fixed radius which, without loss of generality, we take to be $\sqrt{N}$. All this means that the signal may very well be uniform in the hyper-cube, for example. Some clarifications on this matter have been added to the revised version of the manuscript (see l. 170-174 of the revision). This is the reason why we decided to focus on spherically distributed signals in the numerical part.
>
> ----------------------
>
> **Question 4**
>
> This is a good suggestion, and we would like to thank the reviewer for this comment, which has allowed us to improve the quality of our manuscript. Actually, by following steps similar to those detailed in Appendices C.1-C.3 to show Theorem 2, we can derive a state evolution result for the Gaussian AMP with spectral initialization. We have added a comment in l. 206-207 of the revision. We now describe in detail the state evolution corresponding to the Gaussian AMP with spectral initialization in the newly created Appendix C.4 contained in the supplementary material of the revision.
>
> Let us finally remark that the simulation results of Section 4 analyze the *fixed point* of the Gaussian AMP algorithm and, therefore, no difference is expected when considering a spectral initialization instead of the initialization of Eq. (7).
>
> ----------------------
>
> **Question 5**
>
> Theorem 1 and 2 provide an exact characterization in the high-dimensional limit of the performance of the Bayes estimator and of an AMP algorithm, respectively. More specifically, Eq. (10) gives the asymptotic MSE of the Bayes estimator sampling from the (mismatched) posterior, and Eq. (18) provides a general state evolution result for AMP which holds for any pseudo-Lipschitz function $\psi$. Following the second question of this reviewer and a comment from reviewer *ggxP*, we have added Corollary 1 which explicitly gives the asymptotic MSE and overlap of the AMP iterates. The proofs of Theorem 1 and 2 are rather technical and, for space reasons, we had to defer them to the appendices. However, we provide a proof sketch for both these results (see l. 249-259 and 289-299 of the revision).
>
> Our theoretical analysis fuels directly the numerical experiments presented in Section 4, where we investigate in detail the resulting phenomenology. Thus, there is a direct and strict connection between the theoretical and the empirical part of our contribution.
>
> If the reviewer thinks it would increase the clarity of our paper, we would be happy to add a paragraph at the end of Section 1.1. In this paragraph, we could summarize the content of the following sections and make the connection between Theorems 1-2 and the simulation results of Section 4 more clear.
>
> ----------------------
>
> **Question 6**
>
> The reviewer is correct: the characterization of the spectral estimators does not appear in the “Main Results” section, because it is not an original contribution of our paper. As explained in l. 222-224, the asymptotic formulas for the MSE and for the overlap of these estimators follow from previous results in literature, which are reviewed in Appendix A, Theorem 4. We have opted not to incorporate such results in the main text for space reasons: our preference was to use the 9 pages to highlight our novel rigorous results and the surprising phenomenology that arises from them.

---

> > ### Comment · Reviewer_MSna · 2022-08-09
> > **Thanks for the response.**
> >
> > Thank the authors for detailed response to my comments. I'll raise the rating to 6.

---

### Author Response · Authors · 2022-08-01
**General comments on the response**

We would like to thank the reviewers for their numerous valuable comments, which have allowed us to improve the quality of our manuscript. We are glad to read their overall positive evaluation of our work: “good match between predictions and true performance levels” (*Rev. ggxP*); “thorough study [...] with in-depth theoretical analysis and interesting empirical findings” (*Rev. RAkD*); “theoretically solid performance guarantees” (*Rev. oMyw*).

We provide separate replies to the four reviewers, in which we address all their points. We have uploaded a revision of the body of the paper and of the supplementary material as well. In our responses, we point to the parts of the paper that have been modified. The numbering of lines, equations and references refer to the revised version, and the main changes are highlighted in blue color. We have tried to keep such changes to a minimum, in consideration of the strict 9-page limit holding also for the revision uploaded at this time.

---

### Meta-Review · Area_Chair_HfGZ · 2022-08-20

**Recommendation:** Accept
**Confidence:** Certain

**Metareview:**

This paper studies precise high-dimensional asymptotics in a simple low-rank matrix estimation problem. When there exists a distributional mismatch between the true noise distribution and the Gaussian noise assumption imposed to run the AMP algorithm, the authors observe,  and formally quantify, the performance gap between the AMP algorithm and the Bayes estimator.  While the model considered in this paper might still be too idealistic (which limits its broader impacts), the paper is well-written and solid. All reviewers recommend acceptance of this paper, and I echo their recommendation.

**Award:**

No

---

### Decision · Program_Chairs · 2022-09-14

Accept